**Data Availability Statement:** This study obtained de-identified data from the National Health Insurance Service (NHIS) and the Korea Disease

# Risk factors for critical COVID-19 illness during Delta- and Omicron-predominant period in Korea; using K-COV-N cohort in the National health insurance service

Kyung-Shin Lee[1]☯*, Min Jin Go[2]☯, Youn Young Choi[3], Min-Kyung Kim[4], Jaehyun Seong[2], Ho Kyung Sung[5], Jaehyun Jeon[4], Hee-Chang Jang[2]*, Myoung-Hee Kim[6]*

1 Public Health Research Institute, National Medical Center, Seoul, Korea, 2 Division of Clinical Research, National Institute of Infectious Diseases, Korea National Institute of Health, Center for Emerging Virus Research, Cheongju, Republic of Korea, 3 Department of Pediatrics, National Medical Center, Seoul, Korea, 4 Division of Infectious Diseases, National Medical Center, Seoul, Korea, 5 National Emergency Medical Center, National Medical Center, Seoul, Korea, 6 Center for Public Health Data Analytics, National Medical Center, Seoul, Korea

☯ These authors contributed equally to this work.
* mhkim@nmc.or.kr (M-HK); haroc153@gmail.com (H-CJ); kslee0116@nmc.or.kr (K-SL)

## Abstract

### Background

This study evaluated the clinical characteristics of patients with COVID-19 in Korea, and examined the relationship between severe COVID-19 cases and underlying health conditions during the Delta (September 20, 2021 to December 4, 2021) and the Omicron (February 20, 2022 to March 31, 2022) predominant period.

### Methods

This study assessed the association between critical COVID-19 illness and various risk factors, including a variety of underlying health conditions, using multiple logistic regression models based on the K-COV-N cohort, a nationwide data of confirmed COVID-19 cases linked with COVID-19 vaccination status and the National Health Insurance claim information.

### Results

We analyzed 137,532 and 8,294,249 cases of COVID-19 infection during the Delta and the Omicron variant dominant periods, respectively. During the Delta as well as the Omicron period, old age ($\geq$80 years) showed the largest effect size among risk factors for critical COVID-19 illness (aOR = 18.08; 95% confidence interval [CI] = 14.71–22.23 for the Delta; aOR = 24.07; 95% CI = 19.03–30.44 for the Omicron period). We found that patients with solid organ transplant (SOT) recipients, unvaccinated, and interstitial lung disease had more than a two-fold increased risk of critical COVID-19 outcomes between the Delta and Omicron periods. However, risk factors such as urban residence, underweight, and

Control and Prevention Agency (KDCA). The authors had no special access privileges and are not authorized to share or provide the data. Information on how to request the database is available at https://nhiss.nhis.or.kr/bd/ab/bdaba021eng.do. To request the database, visit https://nhiss.nhis.or.kr.

**Funding:** The author(s) received no specific funding for this work.

**Competing interests:** NO authors have competing interests.

underlying medical conditions, including chronic cardiac diseases, immunodeficiency, and mental disorders, had different effects on the development of critical COVID-19 illness between the Delta and Omicron periods.

## Conclusion

We found that the severity of COVID-19 infection was much higher for the Delta variant than for the Omicron. Although the Delta and the Omicron variant shared many risk factors for critical illness, several risk factors were found to have different effects on the development of critical COVID-19 illness between those two variants. Close monitoring of a wide range of risk factors for critical illness is warranted as new variants continue to emerge during the pandemic.

## Introduction

Coronavirus disease 2019 (COVID-19) is a respiratory infection caused by the severe acute respiratory syndrome coronavirus 2 (SARS-CoV-2) [1]. Since its first outbreak at the end of 2019, COVID-19 has spread worldwide, resulting in over six million deaths [1,2]. Over the past three years, the development of effective vaccines and medications has significantly reduced fatality rates of COVID-19, however, the burden of the disease remains higher than that of influenza or other community-acquired respiratory infections [3].

The first confirmed case of COVID-19 in South Korea (hereafter, Korea) was reported on January 20, 2020. The infection spread nationwide, and the number of cases increased rapidly owing to the emergence and spread of several SARS-CoV-2 variants, such as Delta (B·1·617·2; July 7, 2021, to January 29, 2022) and Omicron (B·1·1·529; January 30, 2022, to April 24, 2022). In particular, 8,294,249 confirmed cases were reported during the Omicron-predominant period. As of January 13, 2023, the cumulative number of confirmed cases was 29,698,043. Furthermore, it has been reported that different types of SARS-CoV-2 variants (Alpha [α; B·1·1·7], Beta [β; B·1·351], Delta [B·1·617·2], and Omicron [B·1·1·529]) have different level of infectivity and virulence. Several reports suggested that the Delta variant could cause higher COVID-19 severity compared to the other variants, based on the higher hospitalization rates and emergency care attendance rates observed in the Delta period [4,5]. Moreover, the Omicron variant showed a shorter incubation period and generation time than other SARS-CoV-2 variants [6]. Although the Omicron variants are considered to be associated with less severe disease than former variants, a Hong Kong study reported that among the unvaccinated group, patients infected with Omicron BA·2·2 and those infected with the Delta variant showed similar fatalities [6].

Until now, several studies have examined the association of various underlying diseases with severe COVID-19 [7–9]. While the protective effect of COVID-19 vaccines on reducing SARS-CoV-2 infection severity has been established, factors such as sex, age, genetic characteristics, presence of specific underlying diseases, vaccination status, therapeutic intervention, as well as viral mutations, were found to influence COVID-19 severity [10–12]. According to the Centers for Disease Control and Prevention (CDC) guidelines in the United States, a higher number of underlying diseases predicts a higher risk of severe illness among patients with COVID-19, especially in those living or working in an environment where they cannot receive health care and people with disabilities [10]. The protective effect of COVID-19 vaccines on reducing SARS-CoV-2 infection severity has also been established [13,14].

A review article by the CDC of the United States reported the following were associated with the development of critical COVID-19 cases among adult patients: asthma, cancer, cerebrovascular disease, chronic kidney disease, chronic lung diseases, chronic liver diseases, cystic fibrosis, diabetes mellitus, disabilities, HIV (Human immunodeficiency virus), mental health conditions, neurologic conditions, obesity, physical inactivity, and so on [10]. A study from Korea reported that risk factors for severe COVID-19 cases included diabetes, hypertension, chronic lower respiratory disease, chronic renal failure, and end-stage renal disease [15]. Another Korean study suggested that a previous history of high blood pressure, diabetes, and mental illness was linked to progressing to a critical illness in the early phase of the pandemic[15]. Shi et al showed that age, mechanical ventilation therapy, highest creatinine > 1.5 mg/dL, and combined bloodstream infection were identified as independent predictors of mortality among total patients during the Delta variant [16]. In addition, underlying cardiovascular diseases such as myocardial infarction, cerebral infarction, arrhythmia, and hypertension among patients over 70 years of age were the most important risk factors for COVID-19-related death [17]. Diseases of the immune system, such as non-alcoholic fatty liver disease and autoimmune rheumatoid disease also appeared to cause death in severe cases of COVID-19 [18,19]. However, most analyses of the effects of underlying diseases on COVID-19 severity were conducted before vaccination was widely available to patients and therapeutic agents for COVID-19 were introduced.

Thus, there is a shortage of evidence that defining risk factors for severe COVID-19 should consider the introduction of vaccines and therapeutic agents and the emergence of new variants [20,21]. This study aims to investigate the clinical characteristics of patients with severe COVID-19 and identify underlying health conditions associated with the development of severe COVID-19 cases during the Delta- and Omicron-predominant periods in Korea.

## Materials and methods

### Data source and study population

We accessed the National Health Insurance Service (NHIS) database (https://nhiss.nhis.or.kr/bd/ab/bdaba000eng.do) in August 2022 and used the de-identified data provided by the NHIS, for the following data: 1) sociodemographic information such as age, gender, disability status, health security type (National Health Insurance vs. Medicaid), insurance premium as a proxy for income, and area of residence, 2) regular health check-up information including smoking history, body mass index, and physical activity, and 3) medical claims incurred from healthcare facilities. Recently, the Korea Disease Control and Prevention Agency and NHIS jointly established the Korea Disease Control and Prevention Agency-COVID-19-NHIS cohort (K-COV-N cohort) database, which includes information on confirmed COVID-19 cases and nationwide vaccination statistics. We merged these datasets to link vaccination status and clinical characteristics of patients with COVID-19. S1 Fig showed the epidemic curves for confirmed cases and critical cases by age group based on the database. In this study, we restricted our sample to adults aged ≥20 years who were diagnosed with COVID-19 during Delta and Omicron-predominant periods. Delta predominant period was set to September 20, 2021, to December 4, 2021, and that for Omicron was set to February 20, 2022, to March 31, 2022, based on the weekly detection rates of greater than 99% for each variant (https://kdca.go.kr/contents.es?mid=a20107040000).

### Definition of outcomes

The severity of COVID-19 cases was defined using the World Health Organization clinical status classification, as follows: (1) no limit of activity; (2) limit of activity but no supplementary

$O_2$ required; (3) need for supplemental $O_2$ with nasal prong; (4) need for supplemental $O_2$ with facial mask; (5) non-invasive ventilator or high-flow $O_2$; (6) need for invasive ventilation; (7) multi-organ failure or need for extracorporeal membrane oxygenation (ECMO) therapy or continuous renal replacement therapy (CRRT); and (8) death (https://www.who.int/publications/i/item/covid-19-therapeutic-trial-synopsis).

This study defined the category (4) as 'severe' and (5) ~ (7) as 'critical'. All clinical outcomes were identified using the codes of medical claims within one month after COVID-19 confirmation. The following Electronic Data Interchange (EDI) codes were used to define the 'severe' group to require $O_2$ supply: M0040, M0581, M0582, M0583, M0586, M0587, and M0588. For the 'critical' group, the following codes were used: M0046 (oxygen supply with high flow nasal cannula), MM360, and MM400 (continuous positive airway pressure)/M5850 (mechanical ventilation)/O1903 (extracorporeal membrane oxygenation)/M5850 (mechanical ventilation), and O7031 (CRRT). The fatality was defined as death within a month after COVID-19 diagnosis. We selected the 30-day period from infection to death, which was used as a reasonable proxy variable for COVID-19 fatalities verified in previous studies [22].

## Underlying health conditions

We aimed to evaluate 10 underlying conditions of interest: obesity, diabetes mellitus, immuno-suppression, chronic kidney disease, chronic neurological disease, chronic cardiac disease, chronic pulmonary disease, chronic liver disease, mental disease, and lack of physical activity, based on population-based previous studies [8,11,15,21,23–31] (S1 Table). In a sensitivity analysis, we estimated three groups of weight status such as obesity(25≤BMI), overweight (23≤BMI<25), and underweight(BMI<18.5) versus normal weight(18.5≤BMI<23) as a reference during the Omicron variants, instead of obesity(25≤BMI) versus normal weight (BMI<25) during the Delta variants [32]. The International Classification of Diseases, 10th revision (ICD-10) codes were used to identify all conditions except diabetes mellitus, obesity, and lack of physical activity. From January 1, 2018, to the COVID-19 diagnosis, the patient was presumed to have contracted the disease if the patient had more than one diagnostic code related to the following disease in the claim data of the hospitalization or outpatient prescription with either principal diagnosis or 1st to 4th additional diagnosis in diagnosis statements defined by the ICD-10 (S2 Table). We collected the health examination data in the NHIS database from 2018 to 2020 to identify obesity (BMI 25 kg/m$^2$ and over; based on WPRO criteria [33]), and lack of physical activity. The lack of physical activity was defined as people who had never experienced mild, moderate, or vigorous-intensity activity between 2018 and 2020. We also used the Charlson comorbidity index (CCI), a scoring method to summarize complex comorbidities based on the diagnoses found in the medical records [34].

## Covariates

Covariates included age group (20–39, 40–59, 60–79, ≥80 years), sex (male or female), COVID-19 vaccination status (none, one dose, two or more doses), region (eight metropolitan cities as urban or nine provinces as rural), degree of disability (none, mild, or severe), and income level (Medicaid, 1st to 6th insurance premium deciles as low income, 7th to 13th as middle, or 14th to 20th as high-income level). We defined the group of confirmed cases who were vaccinated within 14 days after 1st or 2nd dose, or within 7 days after vaccination for 3rd dose, were not defined as the vaccinated groups because immunity was considered to be indeterminate.

## Statistical analysis

The descriptive statistics are presented as counts and percentages for categorical variables and median and interquartile ranges for continuous variables to compare the severity scale and demographic characteristics between the Delta and Omicron periods. The $\chi^2$ test was used for categorical variables, and the t-test was performed for continuous variables to test the significance for the comparison of two periods. Associations between risk factors and the development of critical COVID-19 illness were identified by performing univariate and multivariate logistic regression analyses. The potential risk factors included age, sex, COVID-19 vaccination status, region, degree of disability, income level, and presence of underlying conditions. The best predictors among potential risk factors were determined using stepwise regression analysis, and odds ratios (ORs) and 95% confidence intervals (CIs) were estimated. We also checked the variance inflation factor to test whether multicollinearity occurs among risk factors. For risk factor classification, a machine learning random forest algorithm was applied. We performed a sensitivity analysis to identify risk factors associated with a severe or critical illness as secondary outcomes in the Delta and Omicron periods. All statistical analyses were performed using the SAS Enterprise Guide 7.1. (SAS Institute, NC, USA). A two-sided *P*-value <0.05 was considered to indicate statistical significance.

## Ethics statement

This study was approved by the institutional review boards of National Medical Center (approval number NMC-2022-06-059) and Korea Disease Control and Prevention Agency (2022-04-06-P-A). The requirement of informed consent from patients was waived by the board owing to the observational nature of the study.

## Results

### Clinical severity of COVID-19 according to mutant variant (Delta versus Omicron)

During the Delta period, the total number of confirmed cases was 137,532, of which 30,754 were admitted to hospitals (22.4%) and 2,577 had a critical illness (1.9%) (Table 1). Meanwhile, we identified 8,294,249 confirmed cases in the Omicron period, from February 20, 2022, to March 31, 2022, of which 459,709 (5.5%) were admitted and 8,186 (0.1%) had critical illness. The highest prevalence of critical illness was 4.9% during the Delta period and 1.5% during the Omicron period. The median time from diagnosis to critical illness was 2 and 6 days in the Delta and Omicron periods, respectively. In contrast, the median interval from diagnosis to death in the Delta period was 13 days, compared to 10 days in the Omicron period.

### Demographic characteristics of study participants

Table 2 shows the overall sociodemographic and clinical characteristics of COVID-19 cases and the number of patients with critical illness during the Delta and Omicron periods. Among confirmed COVID-19 cases, patients aged over 80 years comprised 5.97% and 4.07% of all cases during the Delta and Omicron periods, respectively. The proportion of patients who received at least two doses of COVID-19 vaccination was 62.84% in the Delta-predominant period, however, 93.92% of patients with COVID-19 were vaccinated twice during the Omicron period. Medicaid beneficiaries comprised 3.97% and 2.56% of all cases during Delta- and Omicron-predominant periods, respectively, while the proportion of patients with an underlying condition was 47.67% and 49.21%.

**Table 1. Severity classification of patients with SARS-CoV-2 infection during Delta and Omicron predominant periods.**

| | Delta-predominant[1] (21.09.20–21.12.04) | | Omicron-predominant[1] (22.02.20–22.03.31) | | P-value |
|---|---|---|---|---|---|
| | N | (%) | N | (%) | |
| Confirmed cases | 137,532 | (100.0) | 8,294,249 | (100.0) | |
| Admission | 30,754 | (22.4) | 459,709 | (5.5) | <0.001 |
| Severe or critical[2] | 9,809 | (7.1) | 76,131 | (0.9) | <0.001 |
| Severe | 7,232 | (5.4) | 67,945 | (0.8) | <0.001 |
| Critical | 2,577 | (1.9) | 8,186 | (0.1) | <0.001 |
| Grade 5 | 2,262 | (1.6) | 6,737 | (0.1) | <0.001 |
| Grade 6 | 190 | (0.1) | 1,328 | (0.02) | |
| Grade 7 | 125 | (0.09) | 121 | (0.001) | |
| Death within 1 month from COVID-19 confirmation | 1,998 | (1.5) | 21,944 | (0.3) | <0.001 |
| Time intervals, from diagnosis to severe condition [median (IQR), days] | 2 (0–5) | | 10 (5–19) | | <0.001 |
| Time intervals, from diagnosis to critical condition [median (IQR), days] | 2 (0–6) | | 6 (3–12) | | <0.001 |
| Time intervals, from diagnosis to death within 1 month [median (IQR), days] | 13 (7–19) | | 10 (5–17) | | <0.001 |

[1] More than 99% for detection rate.

[2] The definition of severe or critical group was as below.

'severe' group to require O2 supply: M0040, M0581, M0582, M0583, M0586, M0587, and M0588. For the 'critical' group, the following codes were used: M0046 (oxygen supply with high flow nasal cannula), MM360 and MM400 (continuous positive airway pressure)/M5850 (mechanical ventilation)/O1903 (extracorporeal membrane oxygenation)/M5850 (mechanical ventilation), and O7031 (CRRT).

Among critical cases, during the Delta and Omicron periods, respectively, the proportion of patients aged ≥80 years were 26.54% and 44.09%; the proportions of Medicaid beneficiaries were 10.52% and 11.83%; and the proportions of those with any underlying conditions were 71.87% and 84.35%. These proportions were consistently higher than those reported among confirmed cases. During the Omicron period, however, the percentage of patients receiving at least two COVID-19 vaccinations was remarkably lower among critically ill patients than in all confirmed cases (59.5% vs. 92.99%).

## Risk factors for critical COVID-19 among adults

Estimated odds ratios (OR) of risk factors during the Delta and Omicron periods from logistic regression models are presented in Table 3. During the Delta period, adjusted ORs (aORs) for the development of critical illness were doubled among patients aged ≥80 years (aOR = 18.08; 95% confidence interval [CI] = 14.71–22.23), those who underwent solid organ or hematopoietic stem cell transplantation (aOR = 2.89; 95% CI = 1.89–4.42), and interstitial lung disease (aOR = 2.08; 95% CI = 1.57–2.76). The patients who had 1st dose (aOR = 0.28; 95% CI = 0.24–0.32), and at least 2nd dose (aOR = 0.13; 95% CI = 0.12–0.14) of vaccination, were protective effect for the development of critical COVID-19 illness compared to unvaccinated patients.

During the Omicron period, we found the highest aOR for the development of critical COVID-19 illness in those aged ≥80 years using the multiple logistic regression model (aOR = 24.07; 95% CI = 19.03–30.44). Solid organ or hematopoietic stem cell transplantation (aOR = 5.85; 95% CI = 4.77–7.17), immunodeficiency (aOR = 2.73; 95% CI = 2.29–3.25), and interstitial lung disease (aOR = 3.10; 95% CI = 2.69–3.58) were related to at least double the aOR than the other risk factors for critical illness.

We found that residence in urban areas, immunodeficiency, myopathies, paralytic syndromes, heart failure and cardiomyopathies, valvular heart disease, arrhythmias, pulmonary

**Table 2. Sociodemographic and clinical characteristics of COVID-19 confirmed cases during Delta and Omicron predominant periods.**

| | Confirmed cases | | | Critical cases | | |
|---|---|---|---|---|---|---|
| | Delta-predominant | Omicron-predominant | *P*-value | Delta-predominant | Omicron-predominant | *P*-value |
| | N (%) | N (%) | | N (%) | N (%) | |
| Total | 137,532 | 8,294,249 | | 2,577 | 8,186 | |
| Sex | | | | | | |
| Male | 71378 (51.90) | 3713372 (44.77) | <0.001 | 1444 (56.03) | 4615 (56.38) | 0.7598 |
| Female | 66154 (48.10) | 4580877 (55.23) | | 1133 (43.97) | 3571 (43.62) | |
| Age (years) | | | | | | |
| 20–39 | 44468 (32.33) | 3159325 (38.09) | <0.001 | 158 (6.13) | 232 (2.83) | <0.001 |
| 40–59 | 43630 (31.72) | 3132271 (37.76) | | 390 (15.13) | 783 (9.57) | |
| 60–79 | 41222 (29.97) | 1664762 (20.07) | | 1345 (52.19) | 3562 (43.51) | |
| 80+ | 8212 (5.97) | 337891 (4.07) | | 684 (26.54) | 3609 (44.09) | |
| Vaccination | | | | | | |
| None | 22207 (16.15) | 396710 (4.78) | <0.001 | 1176 (45.63) | 3014 (36.82) | <0.001 |
| 1st Dose | 28903 (21.02) | 107516 (1.30) | | 257 (9.97) | 259 (3.13) | |
| >2nd Dose | 86422 (62.84) | 7790023 (93.92) | | 1144 (44.39) | 4916 (60.05) | |
| Region | | | | | | |
| Urban | 60127 (43.72) | 4394573 (52.98) | <0.001 | 1160 (45.01) | 4299 (52.52) | <0.001 |
| Rural | 74498 (54.17) | 3864245 (46.59) | | 1404 (54.48) | 3866 (47.23) | |
| Unknown | 2907 (2.11) | 35431 (0.43) | | 13 (0.50) | 21 (0.26) | |
| Degree of disability | | | | | | |
| No | 128267 (93.26) | 7896280 (95.20) | <0.001 | 2030 (78.77) | 5665 (69.20) | <0.001 |
| Mild | 5691 (4.14) | 240156 (2.90) | | 311 (12.07) | 1240 (15.15) | |
| Severe | 3574 (2.60) | 157813 (1.90) | | 236 (9.16) | 1281 (15.65) | |
| Income level | | | | | | |
| Medicaid | 5461 (3.97) | 211925 (2.56) | <0.001 | 271 (10.52) | 968 (11.83) | 0.0006 |
| T1(Poor) | 33260 (24.18) | 1877754 (22.64) | | 557 (21.61) | 1713 (20.93) | |
| T2 | 44017 (32.00) | 2524841 (30.44) | | 695 (26.97) | 1900 (23.21) | |
| T3 (Rich) | 49739 (36.17) | 3464875 (41.77) | | 1014 (39.35) | 3479 (42.50) | |
| Unknown | 5055 (3.68) | 214854 (2.59) | | 40 (1.55) | 126 (1.54) | |
| Underlying Condition | | | | | | |
| Any | 65556 (47.67) | 4081495 (49.21) | <0.001 | 1852 (71.87) | 6905 (84.35) | <0.001 |
| Obesity | 35817 (26.04) | 2208108 (26.62) | <0.001 | 745 (28.91) | 1579 (19.29) | <0.001 |
| Diabetes mellitus | 30657 (22.29) | 1460631 (17.61) | <0.001 | 1279 (49.63) | 4829 (58.99) | 0.4118 |
| Immunosuppression | 18044 (13.12) | 1044966 (12.6) | <0.001 | 651 (25.26) | 3113 (38.03) | <0.001 |
| Cancer | 8628 (6.27) | 476668 (5.75) | <0.001 | 371 (14.4) | 2041 (24.93) | <0.001 |
| Solid organ or hematopoietic stem cell transplantation | 286 (0.21) | 11786 (0.14) | <0.001 | 32 (1.24) | 257 (3.14) | <0.001 |
| Autoimmune disease | 10202 (7.42) | 615891 (7.43) | 0.915 | 313 (12.15) | 1228 (15.00) | 0.0003 |
| Immunodeficiency | 581 (0.42) | 28986 (0.35) | <0.001 | 41 (1.59) | 288 (3.52) | <0.001 |
| Chronic kidney disease | 4077 (2.96) | 156325 (1.88) | <0.001 | 318 (12.34) | 1726 (21.08) | <0.001 |
| Chronic neurological disease | 7632 (5.55) | 303213 (3.66) | <0.001 | 550 (21.34) | 3023 (36.93) | <0.001 |
| Dementia | 4161 (3.03) | 145231 (1.75) | <0.001 | 348 (13.5) | 1748 (21.35) | <0.001 |
| Cerebrovascular disease | 2899 (2.11) | 118062 (1.42) | <0.001 | 203 (7.88) | 1251 (15.28) | <0.001 |
| Myopathies | 397 (0.29) | 22115 (0.27) | 0.1164 | 19 (0.74) | 113 (1.38) | 0.0097 |
| Paralytic syndromes | 2288 (1.66) | 99017 (1.19) | <0.001 | 190 (7.37) | 1137 (13.89) | <0.001 |
| Chronic cardiac disease | 47634 (34.63) | 2223073 (26.8) | <0.001 | 1850(71.79) | 6844(83.61) | <0.001 |

(*Continued*)

**Table 2.** (Continued)

| | Confirmed cases | | | Critical cases | | |
|---|---|---|---|---|---|---|
| | Delta-predominant | Omicron-predominant | *P*-value | Delta-predominant | Omicron-predominant | *P*-value |
| | N (%) | N (%) | | N (%) | N (%) | |
| Coronary artery disease | 4246 (3.09) | 181181(2.18) | <0.001 | 282(10.94) | 1173(14.33) | <0.001 |
| Heart failure and cardiomyopathies | 8585 (6.24) | 358800 (4.33) | <0.001 | 550(21.34) | 2831(34.58) | <0.001 |
| Valvular heart disease | 949 (0.69) | 45266 (0.55) | <0.001 | 46(1.79) | 349(4.26) | <0.001 |
| Arrhythmias | 7319 (5.32) | 358674 (4.32) | <0.001 | 357(13.85) | 1818(22.21) | <0.001 |
| Hypertension | 44204 (32.14) | 2022042 (24.38) | <0.001 | 1769(68.65) | 6589(80.49) | <0.001 |
| Chronic pulmonary disease | 34097 (24.79) | 2868022 (34.58) | <0.001 | 1052(40.82) | 4434(54.17) | <0.001 |
| Chronic obstructive pulmonary disease | 47868 (34.80) | 3107746(37.47) | <0.001 | 1318(51.14) | 5266(64.33) | <0.001 |
| Asthma | 28042 (20.39) | 1838827 (22.17) | <0.001 | 775 (30.07) | 3091 (37.76) | <0.001 |
| Interstitial lung disease | 637 (0.46) | 26000 (0.31) | <0.001 | 65 (2.52) | 403 (4.92) | <0.001 |
| Bronchiectasis | 1638 (1.19) | 73799 (0.89) | <0.001 | 78 (3.03) | 371 (4.53) | 0.0009 |
| Smoking history (current smoking in 2018–2020) | 103 (0.07) | 939088 (11.32) | <0.001 | 1 (0.04) | 495 (6.05) | <0.001 |
| Pulmonary tuberculosis | 340 (0.25) | 14488 (0.17) | <0.001 | 18 (0.7) | 128 (1.56) | 0.0009 |
| Long-term oxygen therapy | 5465 (3.97) | 288102 (3.47) | <0.001 | 242 (9.39) | 984 (12.02) | 0.0002 |
| Chronic liver disease | 19551 (14.22) | 1079673 (13.02) | <0.001 | 546 (21.19) | 1668 (20.38) | 0.3743 |
| Cirrhosis | 1117 (0.81) | 43974 (0.53) | <0.001 | 65 (2.52) | 301 (3.68) | 0.0048 |
| Non-alcoholic fatty liver disease | 15991 (11.63) | 919024 (11.08) | <0.001 | 429 (16.65) | 1203 (14.7) | 0.0160 |
| Alcoholic liver disease | 4016 (2.92) | 187748 (2.26) | <0.001 | 129 (5.01) | 424 (5.18) | 0.7275 |
| Autoimmune hepatitis | 157 (0.11) | 8880 (0.11) | 0.4252 | 4 (0.16) | 36 (0.44) | 0.0384 |
| Mental disorder | 22111 (16.08) | 1186219 (14.30) | <0.001 | 881 (34.19) | 4091 (49.98) | <0.001 |
| Psychotic disorder | 3012 (2.19) | 125414 (1.51) | <0.001 | 142 (5.51) | 849 (10.37) | <0.001 |
| Mood disorder | 21484 (15.62) | 1162761 (14.02) | <0.001 | 862 (33.5) | 3990 (48.7) | <0.001 |
| Lack of physical activity | 76290 (55.47) | 4097666 (49.40) | <0.001 | 1728 (67.05) | 6023 (73.58) | <0.001 |
| Charlson comorbidity index in 2020 (Continuous) | 1.05 (1.71) | 0.84 (1.47) | <0.001 | 2.57 (2.68) | 3.45 (2.86) | <0.001 |
| Number of underlying conditions (Continuous)[1] | 1.60 (1.61) | 1.51 (1.51) | <0.001 | 3.05 (1.83) | 3.82 (1.69) | <0.001 |

[1] Number of diagnosed underlying diseases including obesity, diabetes mellitus, immunosuppression, chronic kidney disease, chronic neurological disease, chronic cardiac disease, chronic pulmonary disease, chronic liver disease, and mental disease.

tuberculosis, cirrhosis, psychotic disorder, and mood disorder were significantly associated with the development of critical illness in the Omicron dominant period, but not in Delta.

The sensitivity analysis using univariate and multivariate logistic regression models showed similar results for severe or critical illness (S3 Table). The random forest model revealed that CCI, age group, and vaccination status were influential risk factors for critical COVID-19 illness during both Delta and Omicron periods (S2 Fig).

## Discussion

COVID-19 is a major threat to global health, with the elderly population and those with underlying health conditions shown to have the highest risk of developing severe COVID-19. However, there is a lack of evidence regarding the association of specific conditions with the development of critical cases with the emergence of new variants and mass vaccinations against COVID-19. To address these changes, our study examined potential risk factors for critical COVID-19 illness during the recent Delta and Omicron predominant period, using the nationwide claims data in Korea.

**Table 3. Parameter estimates from multiple logistic regression models for development of critical condition during Delta and Omicron pre-dominant periods.**

| | Delta-predominant (9/21/20–12/21/04) | | Omicron-predominant (2/22/20–3/22/31) | | Scale of OR with color |
|---|---|---|---|---|---|
| | Univariate model | Multivariate model[1] | Univariate model | Multivariate model[1] | NS (Non-significant) |
| | OR (95% CI) | aOR (95% CI) | OR (95% CI) | aOR (95% CI) | <1·0 |
| Sex | | | | | 1·0–5·0 |
| Male | 1 | 1 | 1 | 1 | 5·1–9·9 |
| Female | 0.71 (0.68, 0.75) | 0.65 (0.60, 0.71) | 0.47 (0.46, 0.48) | 0.49 (0.46, 0.53) | 10·0≤ |
| Age categories, year | | | | | |
| 20–39 | 1 | 1 | 1 | 1 | |
| 40–59 | 2.53 (2.10, 3.04) | 2.75 (2.27, 3.32) | 3.40 (2.94, 3.94) | 2.35 (1.86, 2.96) | |
| 60–79 | 9.46 (8.02, 11.16) | 9.94 (8.27, 11.95) | 29.18 (25.55, 33.32) | 10.45 (8.34, 13.08) | |
| 80+ | 25.48 (21.40, 30.35) | 18.08 (14.71, 22.23) | 146.92 (128.66, 167.78) | 24.07 (19.03, 30.44) | |
| Vaccination | | | | | |
| None | 1 | 1 | 1 | 1 | |
| 1st Dose | 0.16 (0.14, 0.18) | 0.28 (0.24, 0.32) | 0.31 (0.28, 0.36) | 0.42 (0.34, 0.51) | |
| ≥2nd Dose | 0.24 (0.22, 0.26) | 0.13 (0.12, 0.14) | 0.08 (0.08, 0.09) | 0.10 (0.09, 0.11) | |
| Region | | | | | |
| Rural | 1 | 1 | 1 | 1 | |
| Urban | 1.02 (0.95, 1.11) | 0.96 (0.88, 1.04) | 0.98 (0.94, 1.02) | 1.10 (1.02, 1.17) | |
| Unknown | 0.23 (0.14, 0.40) | 0.39 (0.22, 0.68) | 0.59 (0.39, 0.91) | - | |
| Type of disability | | | | | |
| No disability | 1 | 1 | 1 | 1 | |
| Mild disability | 3.60 (3.18, 4.06) | 1.24 (1.08, 1.42) | 7.23 (6.80, 7.69) | 1.18 (1.07, 1.30) | |
| Severe disability | 4.40 (3.83, 5.06) | 1.69 (1.44, 1.97) | 11.40 (10.73, 12.11) | 1.94 (1.72, 2.18) | |
| Income level in 2020 | | | | | |
| Medicaid | 1 | 1 | 1 | - | |
| T1(Poor) | 0.33 (0.28, 0.38) | - | 0.20 (0.18, 0.22) | - | |
| T2 | 0.31 (0.27, 0.36) | - | 0.16 (0.15, 0.18) | - | |
| T3 (Rich) | 0.40 (0.35, 0.46) | - | 0.22 (0.20, 0.24) | - | |
| Unknown | 0.15 (0.11, 0.21) | - | 0.13 (0.11, 0.15) | - | |
| Underlying Condition | | | | | |
| Any | 2.86 (2.62, 3.12) | NA[2] | 5.57 (5.25, 5.91) | NA[2] | |
| Underweight(-18.5) | NA | NA | 5.95 (4.55, 7.79) | 1.79 (1.36, 2.35) | |
| Overweight(23–25) | NA | NA | 1.22 (0.82, 1.83) | 0.93 (0.62, 1.40) | |
| Obesity(25+) | 1.16 (1.06, 1.26) | 1.45 (1.31, 1.60) | 1.90 (1.35, 2.66) | 1.47 (1.05, 2.06) | |
| Diabetes mellitus | 3.54 (3.27, 3.83) | 1.33 (1.21, 1.46) | 6.75 (6.46, 7.05) | 1.38 (1.28, 1.49) | |
| Immunosuppression | 2.29 (2.09, 2.50) | NA[2] | 4.27 (4.08, 4.46) | NA[2] | |
| Cancer | 2.58 (2.31, 2.89) | 1.17 (1.03, 1.32) | 5.47 (5.20, 5.75) | 1.89 (1.75, 2.05) | |
| Solid organ or hematopoietic stem cell transplantation | 6.67 (4.61, 9.65) | 2.89 (1.89, 4.42) | 23.27 (20.52, 26.38) | 5.85 (4.77, 7.17) | |
| Autoimmune disease | 1.75 (1.55, 1.97) | - | 2.20 (2.07, 2.34) | 1.13 (1.03, 1.24) | |
| Immunodeficiency | 4.03 (2.92, 5.54) | - | 10.49 (9.32, 11.81) | 2.73 (2.29, 3.25) | |
| Chronic kidney disease | 4.91 (4.35, 5.55) | 1.35 (1.18, 1.56) | 14.05 (13.32, 14.82) | 1.86 (1.70, 2.04) | |
| Chronic neurological disease | 4.90 (4.45, 5.40) | NA[2] | 15.58 (14.89, 16.30) | NA[2] | |
| Dementia | 5.37 (4.78, 6.04) | 1.24 (1.08, 1.43) | 15.41 (14.61, 16.25) | 1.40 (1.27, 1.56) | |
| Cerebrovascular disease | 4.20 (3.62, 4.87) | - | 12.62 (11.88, 13.41) | - | |
| Myopathies | 2.65 (1.67, 4.20) | - | 5.26 (4.37, 6.34) | 1.62 (1.21, 2.18) | |

*(Continued)*

**Table 3.** (Continued)

| | Delta-predominant (9/21/20–12/21/04) | | Omicron-predominant (2/22/20–3/22/31) | | Scale of OR with color |
|---|---|---|---|---|---|
| | Univariate model | Multivariate model[1] | Univariate model | Multivariate model[1] | NS (Non-significant) |
| | OR (95% CI) | aOR (95% CI) | OR (95% CI) | aOR (95% CI) | <1·0 |
| Paralytic syndromes | 5.04 (4.32, 5.88) | - | 13.49 (12.67, 14.37) | 1.70 (1.51, 1.91) | |
| Chronic cardiac disease | 4.96 (4.55, 5.40) | NA[2] | 13.97 (13.17, 14.81) | NA[2] | |
| Coronary artery disease | 4.06 (3.57, 4.61) | 1.29 (1.12, 1.48) | 7.54 (7.08, 8.02) | 1.12 (1.02, 1.24) | |
| Heart failure and cardiomyopathies | 4.29 (3.89, 4.72) | - | 11.78 (11.25, 12.33) | 1.38 (1.27, 1.50) | |
| Valvular heart disease | 2.70 (2.00, 3.64) | - | 8.17 (7.34, 9.10) | 1.36 (1.14, 1.61) | |
| Arrhythmias | 2.96 (2.64, 3.32) | - | 6.35 (6.02, 6.69) | 1.21 (1.11, 1.32) | |
| Hypertension | 4.77 (4.39, 5.19) | 1.51 (1.36, 1.68) | 12.84 (12.15, 13.56) | 1.77 (1.61, 1.94) | |
| Chronic pulmonary disease | 2.13 (1.97, 2.30) | NA[2] | 2.24 (2.14, 2.34) | NA[2] | |
| Chronic obstructive pulmonary disease | 1.99 (1.84, 2.15) | 1.18 (1.08, 1.29) | 3.01 (2.88, 3.15) | 1.26 (1.16, 1.35) | |
| Asthma | 1.70 (1.56, 1.85) | 1.11 (1.01, 1.22) | 2.13 (2.04, 2.23) | 1.16 (1.08, 1.24) | |
| Interstitial lung disease | 6.08 (4.69, 7.88) | 2.08 (1.57, 2.76) | 16.71 (15.11, 18.48) | 3.10 (2.69, 3.58) | |
| Bronchiectasis | 2.67 (2.12, 3.36) | - | 5.32 (4.79, 5.90) | - | |
| Smoking history (current smoking in 2018–2020) | 0.51 (0.07, 3.68) | - | 0.50 (0.46, 0.55) | - | |
| Pulmonary tuberculosis | 2.94 (1.83, 4.74) | - | 9.15 (7.68, 10.91) | 1.18 (1.01, 1.37) | |
| Long-term oxygen therapy | 2.57 (2.25, 2.95) | - | 3.81 (3.56, 4.07) | - | |
| Chronic liver disease | 1.64 (1.49, 1.81) | NA[2] | 1.71 (1.62, 1.81) | NA[2] | |
| Cirrhosis | 3.29 (2.56, 4.25) | - | 7.21 (6.42, 8.09) | 1.26 (1.05, 1.52) | |
| Non-alcoholic fatty liver disease | 1.53 (1.38, 1.70) | - | 1.38 (1.30, 1.47) | 0.86 (0.79, 0.93) | |
| Alcoholic liver disease | 1.78 (1.48, 2.13) | - | 2.36 (2.14, 2.60) | - | |
| Autoimmune hepatitis | 1.37 (0.51, 3.70) | - | 4.14 (2.98, 5.75) | - | |
| Mental disease | 2.78 (2.56, 3.02) | NA[2] | 6.00 (5.75, 6.27) | NA[2] | |
| Psychotic disorder | 2.68 (2.26, 3.19) | - | 7.58 (7.06, 8.14) | 1.58 (1.39, 1.80) | |
| Mood disorder | 2.79 (2.56, 3.03) | - | 5.85 (5.60, 6.11) | 1.45 (1.34, 1.56) | |
| Lack of physical activity | 1.64 (1.52, 1.79) | 1.36 (1.23, 1.50) | 2.85 (2.72, 3.00) | 1.29 (1.20, 1.38) | |
| Charlson comorbidity index in 2020 (Continuous) | 1.56 (1.53, 1.59) | NA[2] | 1.96 (1.94, 1.98) | NA[2] | |
| Number of underlying disease (Continuous)[3] | 1.49 (1.46, 1.51) | NA[2] | 1.89 (1.87, 1.91) | NA[2] | |

Note: OR, odds ratio; aOR, adjusted odds ratio; CI, confidence interval; NS, Non-significant; NA, Not analysis.

[1] After selecting variables using stepwise regression analysis.

[2] These variables weren't included in stepwise regression model to overcome multicollinearity.

[3] Have diagnosed and number of underlying diseases including obesity, diabetes mellitus, immunosuppression, chronic kidney disease, chronic neurological disease, chronic cardiac disease, chronic pulmonary disease, chronic liver disease, and mental disease.

Our results suggested that there were risk factors for critical illness, including old age, disability, several underlying conditions such as a solid organ or hematopoietic stem cell transplantation, unvaccination, and interstitial lung disease during both Delta and Omicrons predominant period. However, it was observed that the impact of urban residence, underweight, and underlying diseases including chronic cardiac diseases, immunodeficiency, and mental disorders on critical illness differed between Delta and Omicron variants.

Several recent studies have reported differences in severity and clinical characteristics among patients between the Delta and Omicron periods [35,36]. Corriero et al. showed

Omicron variant had a lower incidence of pulmonary embolism and a higher number of comorbidities than the Delta variant [35]. In a retrospective case-control study conducted in Japan, the number of pneumonic patients requiring oxygen support, and patients who needed admission were significantly lower in the Omicron group [36].

We observed a higher risk for critical illness among the elderly population during the Omicron period compared to the Delta period. Although the Omicron variant was less virulent [37], resulting in lower hospitalization and mortality rates, our study showed a stronger effect of mild virulence in the 20–39 age group during the Omicron period, resulting in a relatively increased risk of critical illness in the elderly.

Mental health emerged as a risk factor during the Omicron period, but not the Delta period. As the number of confirmed cases increased, accessing medical care may become more challenging during this period. This difficulty may be particularly notable among confirmed cases with psychiatric disorders. Consequently, it is possible that the symptoms worsened due to delays in receiving timely medical care. Further detailed analysis is needed, considering specific policies targeting mentally ill patients infected with COVID-19.

We found that the time interval from diagnosis to severe or critical illness was longer during Omicron variants compared to Delta variants. This indicator reflects not only the characteristics of the variant itself such as virulence [37], but also the healthcare system contexts. The shorter period from diagnosis to developing a critical illness, which is less than six days during the Delta period, seems to be associated with higher virulence of the Delta variant. However, the time interval from diagnosis to death during the Omicron dominant period was shorter than that during the Delta period. This is likely because the proportion of critical cases in the population over 80 was significantly higher during the Omicron period (44.09%) than during the Delta period (26.54%), and those elderly patients in a critical condition rapidly deteriorated partly due to underlying health conditions and incapacity of healthcare system overwhelmed by the Omicron epidemic, in spite of the low virulence of the Omicron variant. Further studies to clarify the reasons should be followed.

Solid organ transplant (SOT) recipients had more than a two-fold increase in risk for severe COVID-19 outcomes, which supports the observations made in previous studies [38–40]. It revealed that the higher the type and dose of immunosuppressants, the more susceptible the transplanted organ, such as the heart or lungs, the higher the risk of serious illness if the patient is older or has an underlying disease [39,40]. Those patients have to be put on immunosuppressants to suppress rejection in transplanted organs and reduce their defense against virus or bacterial infections [41]. Therefore, SOT patients are at high risk of rapidly multiplying the virus and developing complications such as pneumonia, acute respiratory distress syndrome (ARDS), and sepsis when infected with COVID-19 [42].

Our findings indicate that immunodeficiency and interstitial lung disease (ILD) conferred an increased risk of severe COVID-19. Recent studies reported that ILD patients had an increased risk of death and hospitalization from COVID-19 [43–46]. However, regular assessment of such underlying conditions during different variant periods is required for a better understanding of SARS-CoV-2 infection to prepare for emerging variants of concern. In addition, clinicians should recognize, and properly manage or educate ILD patients about the increased risk of COVID-19 during this pandemic.

The association between pulmonary tuberculosis (TB) and the severity of COVID-19 is not yet fully understood, with conflicting findings from different studies [47–50]. A rapid systematic review and meta-analysis found that tuberculosis was associated with a 2.10-fold increased risk of severe COVID-19 disease, although the statistical difference was not significant [51]. However, other studies have reported mixed findings, with some suggesting an increased risk of mortality in patients with underlying tuberculosis [52], while others indicate that patients

with TB may have a less severe course of COVID-19 and a lower risk of mortality [53]. A study comparing the clinical characteristics of patients with COVID-19/TB and patients with COVID-19 only found that female patients with TB were more likely to have severe COVID-19 [49]. On the other hand, a population-based, dynamic cohort study using national insurance claims databases found that having COVID-19 pneumonia was strongly associated with a higher hazard of detectable active pulmonary tuberculosis [54]. Our study found that the risk of severe illness was significantly increased in cases with TB during the Omicron period (Odds Ratio: 1.18 [95% CI: 1.01, 1.37]), but not during the Delta period. Further research, including larger multicenter studies, is needed to determine the factors influencing mortality and severity in the COVID-19/TB cohort.

Moreover, there is consistent evidence showing that high BMI has been significantly associated with severe COVID-19 [55,56]. Possible mechanisms for the adverse outcome of COVID-19 in obese people include accumulated fat around the airways, which could reduce functional lung capacity, obesity-related pro-inflammatory state which may worsen the pathology of COVID-19, higher viral load and amplified viral shedding that could affect recovery time, as well as hyperinsulinemia and hyperleptinemia that can impair T cell function [57]. Higher BMI has been reported to be associated with severe COVID-19 illness in many countries including Asia [58]. Meanwhile, an England community-based cohort study reported that underweight as well as obese patients were more likely to be hospitalized than patients of healthy weight [57], which is consistent with our findings. Being underweight seems to be associated with frailty and general weakness, which might contribute to the aggravation of COVID-cases, but the mechanism needs to be investigated further.

Our study had some limitations that need to be considered when interpreting the results. Firstly, we defined underlying health conditions using administrative claims data generated only when patients visited healthcare facilities. Although claims data can be useful for certain types of healthcare research, it is important to be aware of its limitations, including potential inaccuracies, lack of clinical detail, and incomplete information on patient outcomes [59]. Consequently, the prevalence in our study would be underestimated. However, the magnitude of underestimation would be minimal given that these conditions are relatively severe and chronic. Second, the insurance claims data did not contain detailed clinical information such as vital signs and laboratory values to provide other important information related to the prognosis of COVID-19. Due to a large sample size, however, it is unlikely that unobserved clinical presentation at the time of admission would seriously skew the findings of the current study. Third, the protective effect of the vaccine against the development of critical illness may be underestimated because we conducted analysis only using confirmed cases. For instance, if the vaccine demonstrated a superior effect in preventing mild COVID-19 cases or infection, the number of vaccinated individuals in the mild patient group would be relatively low, potentially leading to an underestimation of the vaccine's protective effect against severe disease. So, a future study with estimating the COVID-19 severity illness in general population instead of confirmed cases should be needed. In addition, the impact of the COVID-19 vaccination time interval between the general public and vulnerable groups, such as high-risk groups or long-term care facilities, could not be considered in our model. Forth, our study design was population-based retrospective cohort design, but we followed up acute COVID-19 severity of illness, it should be interpreted with caution. It is important to note that longitudinal follow-up studies are required to understand the risk factors for COVID-19 severity according to each variant. Finally, we were unable to account for the effects of therapeutics such as Paxlovid and Molnupiravir [60] because their prescription information was not included in the claims data. Future studies are required to account for such treatment interventions to compensate for the limitations of this study.

The strength of our study is that we used nationwide data to predict the risk factors of critical COVID-19 illness during recent Delta and Omicron predominant periods, considering the vaccination status of the patients. Our findings showed that the effects of risk factors were likely to change for each variant, which warrants further investigation of risk factors to establish robust evidence-based clinical decision-making in future pandemics.

## Conclusion

In conclusion, we found that the absolute number of severe COVID-19 cases was much larger during the Omicron period compared to the Delta period, but the proportion of severe cases among confirmed cases was higher during the Delta period. Several risk factors such as vaccination, income level, and underlying disease including mental disorders differentially affected the development of critical COVID-19 illness between Delta and Omicron period. Close monitoring of a wide range of risk factors for critical illness is warranted as new variants continuously emerge during the pandemic.

## Supporting information

**S1 Fig. Weekly epicurves stratified by age group for confirmed and critical COVID-19 cases.**
(DOCX)

**S2 Fig. Importance ranking of the characteristics of critical patients with COVID-19, according to the Gini index using random forest method.**
(DOCX)

**S1 Table. List of population-based previous studies to investigate risk factors for COVID-19 severity.**
(DOCX)

**S2 Table. ICD-10 Codes used to identify underlying conditions.**
(DOCX)

**S3 Table. Result of logistic regression analysis for severe or critical COVID-19 illness.**
(DOCX)

## Acknowledgments

This study was conducted as part of the public-private joint research on the COVID-19 co-hosted by the KDCA and the NHIS. This study used the database of the KDCA (Korea Disease Control and Prevention Agency, Republic of Korea) and the NHIS (National Health Insurance Service, Republic of Korea) for policy and academic research. The research numbers of this study are KDCA-NHIS-2022-1-527.

## Author Contributions

**Conceptualization:** Kyung-Shin Lee, Min Jin Go, Hee-Chang Jang, Myoung-Hee Kim.

**Formal analysis:** Kyung-Shin Lee, Min Jin Go.

**Investigation:** Youn Young Choi, Min-Kyung Kim, Jaehyun Seong, Ho Kyung Sung, Jaehyun Jeon.

**Methodology:** Ho Kyung Sung.

**Project administration:** Myoung-Hee Kim.

**Supervision:** Hee-Chang Jang, Myoung-Hee Kim.

**Writing – original draft:** Kyung-Shin Lee, Min Jin Go.

**Writing – review & editing:** Youn Young Choi, Min-Kyung Kim, Jaehyun Seong, Ho Kyung Sung, Jaehyun Jeon, Hee-Chang Jang, Myoung-Hee Kim.

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
