## [Decision Letter · Decision Letter 0]

13 Nov 2023

PONE-D-23-23867Risk factors for critical COVID-19 illness during Delta- and Omicron-predominant period in Korea; using K-COV-N cohort in the National health insurance servicePLOS ONE

Dear Dr. Kim,

Thank you for submitting your manuscript to PLOS ONE. After careful consideration, we feel that it has merit but does not fully meet PLOS ONE’s publication criteria as it currently stands. Therefore, we invite you to submit a revised version of the manuscript that addresses the points raised during the review process.

Please revise.

We look forward to receiving your revised manuscript.

Kind regards,

Academic Editor

PLOS ONE

Reviewers' comments:

Reviewer's Responses to Questions

**Comments to the Author**

1. Is the manuscript technically sound, and do the data support the conclusions?

Reviewer #1: Partly

Reviewer #2: Yes

2. Has the statistical analysis been performed appropriately and rigorously? 

Reviewer #1: Yes

Reviewer #2: Yes

3. Have the authors made all data underlying the findings in their manuscript fully available?

Reviewer #1: Yes

Reviewer #2: Yes

4. Is the manuscript presented in an intelligible fashion and written in standard English?

Reviewer #1: Yes

Reviewer #2: Yes

5. Review Comments to the Author

Reviewer #1: Since the therapy and vaccine coverage were significantly different throughout the Delta and Omicron periods, it is very challenging to compare the results of these two time periods.

Please think about including the subject count in the abstract since some readers may only want a quick summary of the study's findings without opening the complete text.

Line 135: An individual was considered to have died from COVID-19 within a month of being diagnosed. Do you have a patient whose stay will be more than one month? What use does having this operational definition serve?

Lines 270–274: The author can explain these data in the discussion section, but it would be better to include them in the results section.

Lines 306-311: The explanation does not fit the assertion..... For those patients, COVID-19 symptoms are exacerbated by uncontrolled and dysregulated immunity (39, 40), and respiratory tract defects may impair the immune response to infection, resulting in susceptibility to SARS-CoV-2, virus spread, pulmonary disease, and systemic inflammation.

The conclusion is not drawn in light of the facts. The severity instances typically occur in the Omicron period, according to Tables 1 and 2.

Reviewer #2: Overall, this study evaluated the clinical characteristics of Covid-19 patients in Korea, and examined the association between severe COVID-19 cases and underlying health conditions during the Delta and Omicron period. The authors also acknowledged some limitations of the study (line 329-353). Thus, I have only minor comments:

1) Line 80: the abbreviation (CDC) should be used because it has been defined in line 75.

2) Table 1: change “from diagnosis death” to “from diagnosis to death”.

3) Table 1, the time interval from diagnosis to death in Omicron is shorter than Delta, although the time interval from diagnosis to severe and critical are shorter in Delta and Omicron. Can the authors elaborate the findings in the discussion section?

4) Table 2 and 3: how about the underlying HIV infections? It is strange that with the large number of data, it can’t capture the underlying HIV status, although it is mentioned in line 83.

5) Line 288 and 289: describing virus with “less toxic” is not common. Please use another term, for example less virulent or less severe. It is also applied for the whole manuscript.

6) In the discussion, I would suggest including discussion about the association between pulmonary tuberculosis (TB) and Covid-19 severity. I think TB is the only infectious disease examined in this study?

6. PLOS authors have the option to publish the peer review history of their article (what does this mean?). If published, this will include your full peer review and any attached files.

Reviewer #1: No

Reviewer #2: **Yes: **Mohamad S. Hakim, PhD.

---

## [Author Response · Author response to Decision Letter 0]

14 Dec 2023

We revised the manuscript according to the reviewers’ valuable and helpful comments, and have provided our point-by-point responses to each comment in the “Review Comments to the Author_231211_f.docx”

---

## [Decision Letter · Decision Letter 1]

26 Feb 2024

Risk factors for critical COVID-19 illness during Delta- and Omicron-predominant period in Korea; using K-COV-N cohort in the National health insurance service

PONE-D-23-23867R1

Dear Dr. Kim,

We’re pleased to inform you that your manuscript has been judged scientifically suitable for publication and will be formally accepted for publication once it meets all outstanding technical requirements.

Kind regards,

Academic Editor

PLOS ONE

Additional Editor Comments (optional):

Reviewers' comments:

Reviewer's Responses to Questions

**Comments to the Author**

1. If the authors have adequately addressed your comments raised in a previous round of review and you feel that this manuscript is now acceptable for publication, you may indicate that here to bypass the “Comments to the Author” section, enter your conflict of interest statement in the “Confidential to Editor” section, and submit your "Accept" recommendation.

Reviewer #2: All comments have been addressed

2. Is the manuscript technically sound, and do the data support the conclusions?

Reviewer #2: Yes

3. Has the statistical analysis been performed appropriately and rigorously? 

Reviewer #2: Yes

4. Have the authors made all data underlying the findings in their manuscript fully available?

Reviewer #2: Yes

5. Is the manuscript presented in an intelligible fashion and written in standard English?

Reviewer #2: Yes

6. Review Comments to the Author

Reviewer #2: The authors have addressed my comments very well. I did not see any concern about this article, including dual publications, research or publication ethics.

7. PLOS authors have the option to publish the peer review history of their article (what does this mean?). If published, this will include your full peer review and any attached files.

Reviewer #2: **Yes: **Mohamad S. Hakim, PhD.

---

## [Editor Report · Acceptance letter]

5 Mar 2024

PONE-D-23-23867R1 

PLOS ONE

Dear Dr. Kim, 

I'm pleased to inform you that your manuscript has been deemed suitable for publication in PLOS ONE. Congratulations! Your manuscript is now being handed over to our production team.

Kind regards, 

on behalf of

Dr. Robert Jeenchen Chen 

Academic Editor

PLOS ONE